

# Comparative analysis of codon usage patterns in chloroplast genomes of five Miscanthus species and related species

Jiajing Sheng[1], Xuan She[2], Xiaoyu Liu[1], Jia Wang[3] and Zhongli Hu[2]

[1] Nantong University, School of Life Sciences, Jiangsu Key Laboratory of Neuroregeneration, Co-innovation Center of Neuroregeneration, Nantong, China
[2] Wuhan University, Wuhan, China
[3] Anhui University of Science and Technology, Huainan, China

## ABSTRACT

Miscanthus is not only a perennial fiber biomass crop, but also valuable breeding resource for its low-nutrient requirements, photosynthetic efficiency and strong adaptability to environment. In the present study, the codon usage patterns of five different *Miscanthus* plants and other two related species were systematically analyzed. The results indicated that the cp genomes of the seven representative species were preference to A/T bases and A/T-ending codons. In addition, 21 common high-frequency codons and 4–11 optimal codons were detected in the seven chloroplast genomes. The results of ENc-plot, PR2-plot and neutrality analysis revealed the codon usage patterns of the seven chloroplast genomes are influenced by multiple factors, in which nature selection is the main influencing factor. Comparative analysis of the codon usage frequencies between the seven representative species and four model organisms suggested that *Arabidopsis thaliana*, *Populus trichocarpa* and *Saccharomyces cerevisiae* could be considered as preferential appropriate exogenous expression receptors. These results might not only provide important reference information for evolutionary analysis, but also shed light on the way to improve the expression efficiency of exogenous gene in transgenic research based on codon optimization.

## INTRODUCTION

*Miscanthus Andersson* (Poaceae) is a C4 photosynthetic plant, which have been widely investigated as a potential second-generation bio-energy crop (*Barling et al., 2013*). The genus Miscanthus includes approximately 20 species, which could be classified into Miscanthus clades and Triarrhena clades (*Ge et al., 2017*). China is the biological diversity center of *Miscanthus* species, of which *Miscanthus lutarioriparius L.Liou* (*M. lutarioriparius*), *Miscanthus sinensis Andersson* (*M. sinensis*), *Miscanthus sacchariflorus (Maxim.) Nakai* (*M. sacchariflorus*) and *Miscanthus floridulus (Lab.) Warb. ex Schum. et Laut* (*M. floridulus*) are the four most widely distributed species. In addition to being bioenergy plants, *Miscanthus* species possess extensive breeding values due to their extremely advantageous agricultural characteristics, such as high photosynthetic efficiency, cold tolerance and an extensive ability to adapt to environmental change

Corresponding author
Zhongli Hu, huzhongli@whu.edu.cn

(*Vermerris, 2008*). Currently, the research focus of *Miscanthus* species is to utilize it as a promising genetic resource (*Clark et al., 2015*; *Zhang et al., 2013*). The more diverse genetic resources available, the more likely it is that scientific research will be able to comprehend the adaptation, evolution and utilization of these significant economic crops.

Chloroplasts (cp) are key plastids involved in multifunctional processes of the plant cell, including photosynthesis, carbon fixation, starch storage, nitrogen metabolism, fatty acid and nucleic acid synthesis (*Jarvis & López-Juez, 2013*; *Nielsen et al., 2016*). Typically, cp genomes (cpDNAs) possess a small size, conserved gene content and large copy numbers, which have been extensively used as valuable source for evolution analysis and plastid engineering (*Amiryousefi, Hyvönen & Poczai, 2018*; *Ravi et al., 2008*; *Yan et al., 2019*). The lack of genomic resources of the *Miscanthus* species has hindered the adequate understanding of their diversity traits (*Chae et al., 2014*; *Sheng et al., 2016*) The low expression efficiency of an exogenous gene may limit the research progress on the functional studies of *Miscanthus* and their related species (*Wu et al., 2021*). Stemming from the rapid development of cp engineering, plasmid DNA have been transferred into the cpDNA of a variety of plants, such as *Nicotiana tabacum*, *Manihot esculenta Crantz* and *Eruca sativa Mill* (*Havaux, Lütz & Grimm, 2003*; *Khodakovskaya et al., 2006*; *Kwak et al., 2019*). Recently, the cpDNA of some *Miscanthus* species have been made available in the National Center for Biotechnology Information (NCBI) database (*Sheng et al., 2021*). These complete cpDNA sequences of *Miscanthus* species can be used for studying population genetics, evolution analysis and plastid engineering (*Amiryousefi, Hyvönen & Poczai, 2018*; *Yan et al., 2019*).

Codon usage bias refers to the variations on the usage frequencies of synonymous codons (*Plotkin & Kudla, 2011*). The pattern of codon usage could be caused by multi-factors during the process of genome and gene evolution, including natural selection, compositional mutation mode, translational selection, gene length, tRNA abundance and mRNA secondary structure (*Liu et al., 2004*; *Pop et al., 2014*; *Quax, Claassens & Söll, 2015*; *Tuller et al., 2010*). The studies of codon preference can not only reveal the evolutionary rules between genes in a species or related species, but also improve the expression efficiency of exogenous sequences in transgenic research by codon optimization. Recently, the applicability of codon optimization in the cpDNA have been determined for many vascular plants, including *Poaceae* (Gramineae) (*Zhang et al., 2012*), *Cinnamomum camphorn (L.) presl* (Camphor tree) (*Chen et al., 2017*), *Fragaria ×ananassa Duch* (strawberry) (*Cheng et al., 2017*) and *Solanum tuberosum L.* (potato) (*Zhang et al., 2018*). However, the codon usage pattern of cpDNA in *Miscanthus* and related species has not been fully elucidated.

In this study, the codon usage patterns of the cpDNA of seven *Miscanthus* and related species, including *M. sinensis*, *Miscanthus transmorrisonensis Hayata* (*M. transmorrisonensis*), *M. floridulus*, *M. sacchariflorus*, *Miscanthus x giganteus* (*M. x giganteus*), *Sorghum bicolor* and *Saccharum spontaneum* were systematically analyzed based on the previously published genome-wide data. Among them, *M. sinensis*, *M. transmorrisonensis* and *M. floridulus* belong to the Miscanthus clades under the *Miscanthus* species, *M. sacchariflorus* belongs to the Triarrhena clades under the *Miscanthus* species, and *M. x giganteus* is a natural triploid hybrid. According to our previous studies,
**Table 1** Genomic features of chloroplast genomes of the seven *Miscanthus* and related species (the total number of amino acids: L_aa ; the GC content at the first, second and third codon positions: GC1, GC2and GC3; average GC at three locations: GC123).

| Parameters | *Miscanthus floridulus* | *Miscanthus giganteus* | *Miscanthus sacchariflorus* | *Miscanthus sinensis* | *Miscanthus transmorrisonensis* | *Saccharum spontaneum* | *Sorghum bicolor* |
|---|---|---|---|---|---|---|---|
| L_aa | 19611 | 19469 | 19508 | 19506 | 19486 | 16553 | 17490 |
| CDSs number (before filting) | 106 | 106 | 122 | 122 | 106 | 76 | 84 |
| CDSs number (after filting) | 65 | 64 | 64 | 64 | 64 | 48 | 52 |
| GC1 | 0.473 | 0.474 | 0.472 | 0.472 | 0.473 | 0.477 | 0.476 |
| GC2 | 0.397 | 0.397 | 0.396 | 0.396 | 0.397 | 0.395 | 0.393 |
| GC3 | 0.312 | 0.311 | 0.311 | 0.311 | 0.311 | 0.302 | 0.303 |
| GC123 | 0.394 | 0.394 | 0.393 | 0.393 | 0.394 | 0.391 | 0.39 |

*Miscanthus* are most closely related to *Sorghum bicolor* (L.) *Moench* and *Saccharum L* (*Sheng et al., 2021*; *Sheng et al., 2017*). In addition, the codon usage bias of these seven species was compared with the other four model species including *Populus trichocarpa Torr & Gray*, *Escherichia coli*, *Arabidopsis thaliana (L.) Heynh* and *Saccharomyces cerevisiae*. *Miscanthus* are not only potential bio-energy crops, but also forms an excellent breeding resources. So, it is of interest to understand the codon usage of *Miscanthus* for better utilization of *Miscanthus* and related resources as germplasm resources. Here, we revealed the codon usage patterns of the *Miscanthus* and related species and determined optimal codons for cpDNA genetic engineering. The results in the current study will not only provide insight into genetic evolution studies, but also provide a reference for selecting appropriate heterologous expression hosts to improve the gene expression of *Miscanthus* plants by optimizing codon.

## MATERIALS & METHODS

### Genomes and sequences selection

The complete cpDNAs of *M. floridulus* (NC_035750.1), *M. sacchariflorus* (NC_028720.1), *M. sinensis* (NC_028721.1), *M. x giganteus* (NC_035753.1), *M. transmorrisonensis* (NC_035752.1), *Sorghum bicolor* (NC_008602), *Saccharum spontaneum* (NC_034802.1) with gene annotation were downloaded from the NCBI GeneBank database. The number of raw sequence coding for amino acids in protein (CDS) of above seven species was 106, 122, 122, 106, 106, 84 and 76 respectively (Table 1). To avoid sampling bias, the CDS sequences were screened from genome-wide data by python scripts (https://github.com/shexuan/codon_analysis) according to the following principles: (1) CDS contains initiation codon (ATG), termination codons (TAA, TAG or TGA) and without intermediate stop codons in the sequences; (2) the number of bases in each CDS must be the fold of three (3) the length of the sequence of CDS should be $\geq 300$ bp (*Wright, 1990*; *Zhang et al. 2007*). After filtration, the CDS number, the base composition at the first/second/third site of codons (GC1/GC2/GC3) and average GC, as well as the total amino acids encoded by each CDS were calculated.

## Analysis of relative synonymous codon usage (RSCU) and relative synonymous codon usage frequency (RFSC)

Relative synonymous codon usage value of a codon (the number of codon occurrences in a gene divided by the number of codon appearances expected under the same codon usage) is the ratio of its actual frequency of utilization to the expected usage frequency without bias. The RSCU was calculated as Eq. (1):

$$\text{RSCU} = \frac{x_{ij}}{\sum_{j}^{N_i} x_{ij}} n_i \qquad (1)$$

where $x_{ij}$ represents the frequency of codon j encoding the i th amino acid, and $n_i$ represents the number of synonymous codon encoding the i th amino acid (*Sharp & Li, 1986*). If the RSCU value of a codon is equal to 1, the codon is used without bias, whereas a RSCU value greater than 1 reflects a significant codon usage bias (*Sharp et al., 1993*).

The RFSC value refers to the proportion of the actually observed number of a codon in the number of all synonymous codons. The RFSC were calculated using Eq. (2):

$$\text{RFSC} = \frac{x_{ij}}{\sum_{j}^{N_i} x_{ij}} \qquad (2)$$

where $x_{ij}$ represents the frequency of codon j encoding for the I th amino acid. The high-frequency codon was screened based on the results of RFSC in all codon. The screening principles were as follows: the RFSC > 60% of one codon; or the RFSC of a codon exceeds the average frequency of synonymous codon by 0.5 times (*Zhou et al., 2007*).

## Determination of optimal codons

The effective number of codons (ENC) can be applied to describe the extent of deviation of codon usage from the random selection, which reflects the degree of unbalanced use of synonymous codon in genes. The ENC value range from 20 (each amino acid uses only one synonymous codon) to 61 (Each synonymous codon is equally used), which is inversely proportional to the codon bias (*Wright, 1990*). The ENC value in each species was calculated by CodonW software and then 10% of the CDS with remarkably high and low expression levels were filtered out according to the ENC value. The RSCU of each codon was obtained from the sequence files of the high and low groups according to the cusp function of emboss (https://www.bioinformatics.nl/emboss-explorer/). Optimal codons were determined by ΔRSCU method. Specifically, the average RSCU values of the two amino acid groups were computed and subtract subsequently (ΔRSCU). The codon will be identified as the optimal codon through comparing the high and low group of the same codon ΔRSCU (>0.08) and RSCU value (high group > 1, low group < 1) (*Romero, Zavala & Musto, 2000*).

## Comparative analysis of codon usage frequency

The ratio of codon usage frequency is one indicator of codon usage bias among species. To further explore the codon usage patterns in the seven species of *Miscanthus* and their relatives, codon usage bias data of four model species including *Escherichia coli* (http://www.kazusa.or.jp/codon/cgi-bin/showcodon.cgi?species=199310); *Saccharomyces*

*cerevisiae* (http://www.kazusa.or.jp/codon/cgi-bin/showcodon.cgi?species=4932); *Populus trichocarpa* (http://www.kazusa.or.jp/codon/cgi-bin/showcodon.cgi?species=3694) and *Arabidopsis thaliana* (http://www.kazusa.or.jp/codon/cgi-bin/showcodon.cgi?species=3702), which have been used as the most common heterologous expression hosts were downloaded from the Codon Usage Database. Subsequently, the codon usage frequencies of the seven species in this study were compared with the above four model organisms. When the ratio is ≥2 or ≤0.5, it suggests that the codon bias difference between the two organisms is significant, whereas other values outside of this range represent a lack of significance (*Pan et al., 2013*).

## Analysis of ENC-plot

The proportion of G and C content at the third position of a codon to the total number of gene bases are defined as GC3s. ENC-plot is plotted with ENC values as ordinate and GC3 value as abscissa, which can be used to analyze the codon usage characteristics of each gene and to explore the relevance between gene base component and codon preference (*Wright, 1990*). ENC values are located on or near the expected curve, when mutation pressure plays a key role in the formation of codon usage patterns. Conversely, when the use of a codon is constrained by natural selection, the ENC value will be well below the prospective curve (*Wright, 1990*).

## PR2-plot analysis

In G3/(G3+C3) as the abscisic and A3/(A3+T3) as the ordinate graphic mapping (PR2-plot), which is performed to explore the composition of the four bases at the third nucleotide position of each codon (*Sueoka, 1999*; *Sueoka, 1995*). The pattern of splashes around the central spot (A = T, C = G) indicate the extent and orientation of the base offset.

## Analysis of Neutrality plot

Neutrality analysis is used to exploring the degree of impact between natural selection and mutation pressure on the mode of codon usage (*Sueoka, 1988*). GC12 indicates the mean GC content at the first and second nucleotide positions of the codon, while GC3 represents the GC content of the third site. GC content at the third nucleotide position of a codon was counted post eliminating the Codon Met (ATG) and Trp (TGG). GC3 was counted post the eliminating of the three stop codons (TAA, TAG and TGA) and three codons (ATT, ATC and ATA) of Ile (*Sueoka, 1988*). Both the GC12 and GC3 values of the seven cpDNAs were counted by Python scripts (https://github.com/shexuan/codon_analysis). If the gradient of the curve regression is 0, indicating that there is no impact of mutation pressure. Gradient 1 represents complete neutrality, which describes that codon usage preference is completely influenced by mutation pressure (*Sueoka, 1988*).

## Correspondence analysis of codon usage

The variations of codon usage in the seven analyzed cpDNAs were investigated based on the correspondence analyses (COA) using CodonW (*Anue et al., 2019*). The usage patterns of 59 codons (exincluding Met, Trp and three termination codons) were compared and all genes can be embedded into a 59-dimension hyperspace, in which each dimension corresponds

to the synonymous codon usage of the gene (*Xiang et al., 2015*). Therefore, the major trends (Axis 1) of these axes in the 59-dimensional hyperspace can be used to determine the maximum fraction of genetic variation, indicating the major sources of codon usage variation. In addition, according to the results of COA, the correlation index between Axis1 and codon usage exponent, including the GC content of codons, GC3s, codon adaptation index (CAI) and the total numbers of amino acids in the encoded polypeptide (L_aa) were computed by python scripts package (https://github.com/shexuan/codon_analysis). CAI value is widely applied to assess gene expression levels, ranging from 0 to 1. Specifically, the larger the CAI value is, the stronger the codon usage preference is, and vice versa (*Sharp & Li, 1986*).

# RESULTS

## Characteristics of codon usage bias
### Analysis of codon base composition
The screened CDSs numbers processed by Python scripts are 65, 64, 64, 64, 64, 48 and 52 for *M. floridulus*, *M. giganteus*, *M. sacchariflorus*, *M. sinensis*, *M. transmorrisonensis*, *Saccharum spontaneum* and *Sorghum bicolor* respectively. In addition, the GC contents of three positions of codons (GC1, GC2, GC3) were calculated respectively (Table 1). It was found that the contents of GC at all three sites and the average GC content (GC123) were all less than 0.5, which indicated the seven analyzed cpDNAs were prone to use A/T bases and A/T-ending codons (Table 1). Specifically, the mean GC content of three sites in *M. floridulus*, *M. giganteus* and *M. transmorrisonensis* was determined to equal the same value (0.375), as well as a matching value determined for *M. sacchariflorus* and *M. sinensis* (0.393), but slightly different in *Saccharum spontaneum* (0.391) and *Sorghum bicolor* (0.39) (Table 1). Furthermore, the distribution trend of GC content was GC1 > GC2 > GC3, indicating that GC was not evenly distributed in the three positions of a codon of the seven assessed species. In summary, the codon usages of GC content in these seven cpDNAs were similar and were biased towards A/T bases.

## RSCU and RFSC
The cp genomes of the seven *Miscanthus* and related species have 30 common codons (RSCU > 1) with 28 codons ending with the nucleotides A/T (93.3%) (Table S1). Therefore, almost the majority of codons of the seven plants species analyzed (RSCU > 1) are likely to end with A/T. The variation ranges in the RSCU values were close in the seven cpDNAs, *i.e.*, 0.31–1.93 in *M. floridulus*, *M. giganteus* and *M. transmorrisonensis*, 0.32–1.94 in *M. sacchariflorus* and *M. sinensis*, 0.32–2.01 in *Saccharum spontaneum* and 0.33–2.04 in *Sorghum bicolor*, respectively (Table S1). In addition, the maximum and the minimum RSCU values belonged to TTA and CTG which encode Leu, indicating the vitally positive bias. Furthermore, the pattern of codon usage were summarized in the seven *Miscanthus* and related species (Fig. 1). Specifically, the high-frequency codons of seven *Miscanthus* and related species possess strong common base and share a total of 21 high-frequency codons (Table S1). Furthemore, *Saccharum spontaneum* and *Sorghum*
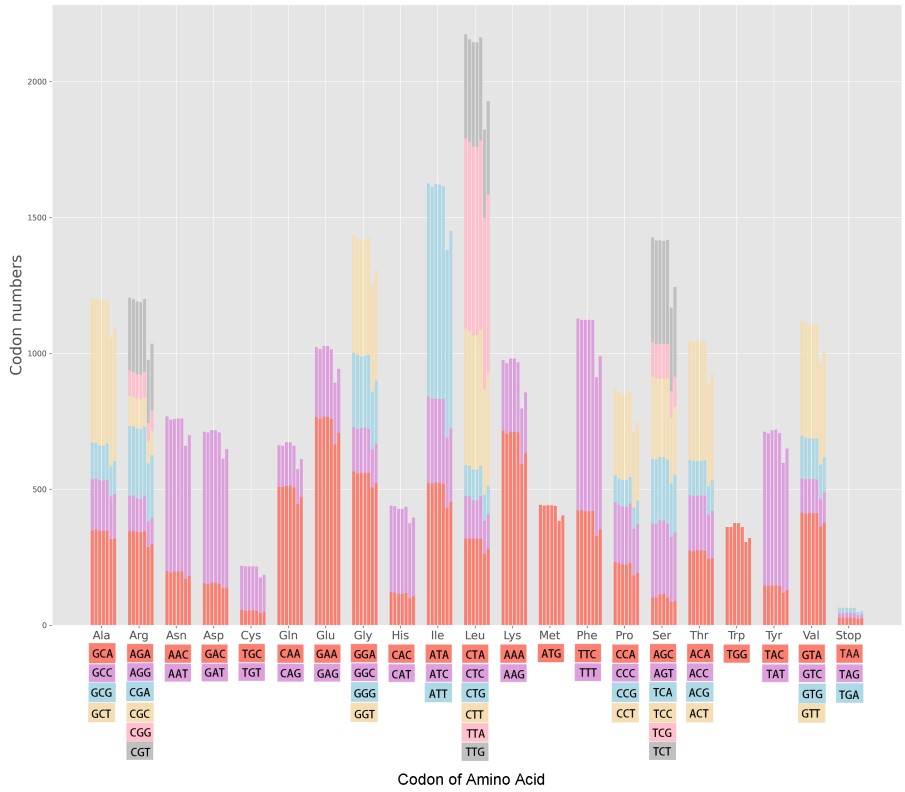

**Figure 1** **Codon content in all protein-coding genes of the seven *Miscanthus* and related cp genomes.** From left to right: *M. floridulus*, *M. giganteus*, *M. sacchariflorus*, *M. sinensis*, *M. transmorrisonensis*, *Saccharum spontaneum* and *Sorghum bicolor*.

*bicolor* were determined to possess two more high-frequency codons, specifically CGT and TAA, than the other five *Miscanthus* species.

## Determination of optimal codons

The ENC values of each CDS were ranked and 10% of genes from both ends were selected to establish high and low expression gene banks respectively. The RSCU values and ΔRSCU values in the two expression libraries were calculated and are listed in Table S2. According to the values of ΔRSCU, the optimal codons in the seven assessed species were determined as follows (Table 2).

## Codon usage frequency

The codon usage frequencies of the seven cpDNAs were compared with four model species including *Escherichia coli*, *Saccharomyces cerevisiae*, *Arabidopsis thaliana* and *Populus trichocarpa* (Table S3). The Results of our analyses indicated that there is little divergence in the codon usage frequencies among the seven assessed plant species with *Saccharomyces cerevisiae*, *Arabidopsis thaliana* and *Populus trichocarpa*, have 9 to11 (accounting for 14.1%–17.2% of total codons), 13 to 14 (20.3%–21.9%), 12 to 13 (18.8%–20.3%) different codons, respectively (Table S3). However, the codon usage frequencies of the seven

**Table 2  Optimal codons in chloroplast genomes of the seven *Miscanthus* and related species.**

| Species | Optimal codon numbers | Optimal codon |
|---|---|---|
| *Miscanthus floridulus* | 4 | 'CAC', 'CCA', 'TCA', 'TAG' |
| *Miscanthus giganteus* | 6 | 'TTC', 'CTT', 'CCA', 'AGG', 'TCA', 'TGA' |
| *Miscanthus sacchariflorus* | 4 | 'GCC', 'CTT', 'AGG', 'ACG' |
| *Miscanthus sinensis* | 11 | 'GCC', 'TTC', 'GGC', 'ATA', 'CTA', 'AGA', 'AGG', 'TCT', 'ACC', 'ACG', 'GTC' |
| *Miscanthus transmorrisonensis* | 4 | 'CTT', 'CCA', 'TCA', 'TGA' |
| *Saccharum spontaneum* | 4 | 'CTT', 'CCA', 'TCA', 'TAG' |
| *Sorghum bicolor* | 8 | 'GCC', 'GGA', 'CAT', 'ATA', 'TCA', 'ACA', 'TAG', 'TGA' |

species with *Escherichia coli* were comparatively higher (27 different codons). The results indicated that the codon frequency difference between *Miscanthus* species and *Arabidopsis thaliana*, *Populus trichocarpa* and *Saccharomyces cerevisiae* was the lowest, while was the largest with *Escherichia coli*. Based on above results, it was optimal to select *Saccharomyces cerevisiae*, *Arabidopsis thaliana* and *Populus trichocarpa* as heterologous expression hosts for *Miscanthus* and related species. Furthermore, the results indicated that TAG is a different termination codon in usage frequency when comparing the seven assessed plants with the four model species (Table S3).

## Source analysis of variation in codon usage
### ENC-plot
The ENC and GC3s of the seven analyzed cpDNAs were plotted. It can be seen from Fig. 2 that the ENC values of most genes were lower than expected values and lie below the standard curve. The results of ENC-plot analysis suggested that codon usage preference of the seven cpDNAs is mainly influenced by natural selection and other factors, while mutation pressure was determined to play only a minor role.

### PR2-plot
PR2-plot is an efficient method to indicate the influence of mutation pressure by investigating the composition of A, T, C and G at the third nucleotide position of a codon. Our results revealed that the AT-bias is 0.464, 0.463, 0.463, 0.463, 0.463, 0.465 and 0.463 for *M. floridulus*, *M. giganteus*, *M. sacchariflorus*, *M. sinensis*, *M. transmorrisonensis*, *Saccharum spontaneumand* and *Sorghum bicolor*, while the GC-bias is 0.512, 0.513, 0.516, 0.515, 0.512, 0.515 and 0.518, respectively (Fig. 3). Therefore, T/G-bias was observed in all seven assessed species. When considered together, codon usage bias of A/T and G/C in the seven cp genomes was unbalanced, indicating that the base composition of the seven analyzed cpDNAs is not only influenced by mutation pressure, but also by natural selection.

### Neutrality plot
The distribution range of GC12 (the mean GC content at the first and second nucleotide positions of a codon) and GC3 is relatively concentrated, in which the range of GC12 is
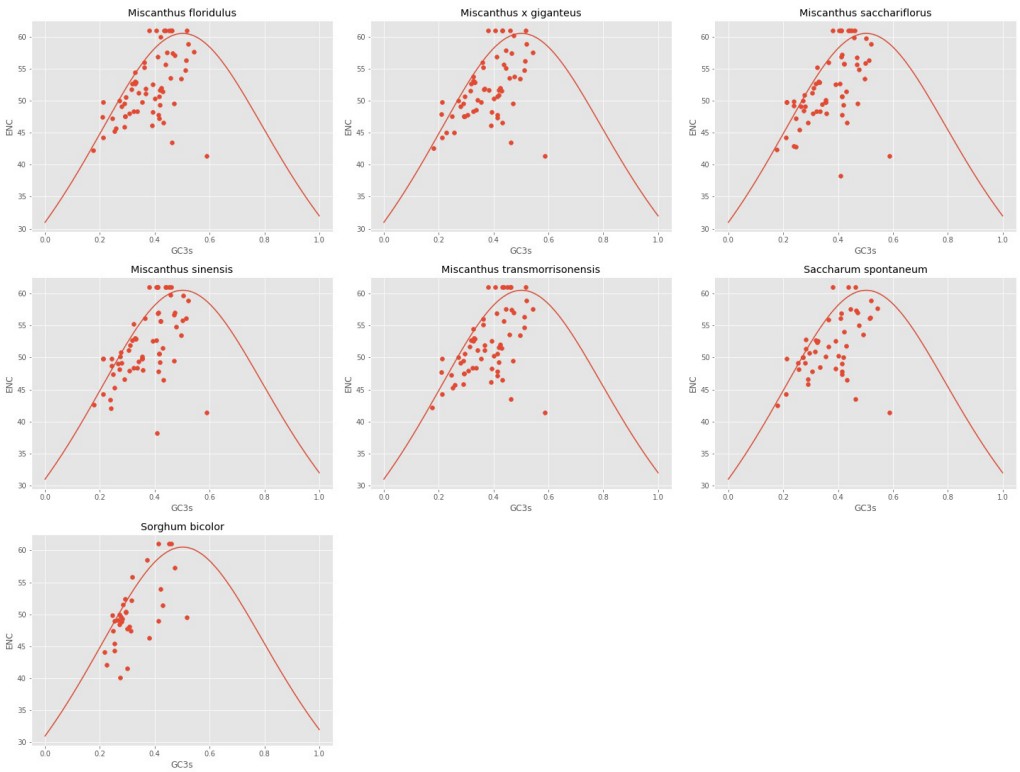

**Figure 2  ENc-plot of chloroplast genomes of seven *Misacanthus* and related species.**

0.3272 to 0.5469, and the range of GC3 is 0.179 to 0.512 (Fig. 4). No significant correlation was found for GC1 with GC2 (r1 = 0.157, r2 = 0.168, r3 = 0.128, r4 = 0.127, r5 = 0.161, r6 = 0.155, r7 = 0.140), GC1 with GC3 (r8 = 0.092, r9 = 0.079, r10 = 0.055, r11 = 0.049, r12 = 0.100, r13 = 0.242, r14 = 0.202) and GC2 with GC3 (r14 = 0.054, r14 = 0.053, r15 = −0.014, r16 = −0.020, r17 = 0.063, r18 = −0.032, r19 = −0.032), which suggested mutation pressure only contributes a minor role in the codon usage preference. In addition, the regression coefficient (slope of neutrality plot) was 0.006 to 0.198, indicating that the correlation between GC12 and GC3 is not significant, and the composition of the first two bases may be different from the third base of the codon. These results demonstrated that the codon usage patterns of cp coding sequences in the seven species are mainly affected by natural selection.

## Correspondence analysis (COA)

Correspondence analysis is used to explore the variations of codon usage among the analyzed cpDNAs. In the current study, RSCU-based COA was used to compare the usage patterns of 59 codons, which produced a series of orthogonal axes, reflecting the trend of change of codon usage in the seven *Miscanhus* and related plant species cpDNAs. The first four axes accounted for 36.2%, 38.2%, 38.5%, 38.3%, 36.8%, 40.7% and 40.9% of the overall changes, while the first axis proportion to 14.3%, 15.9%, 15.8%, 15.8%, 14.6%, 15.6% and 9.7% of the total variation in seven species respectively (Table 3). Axis
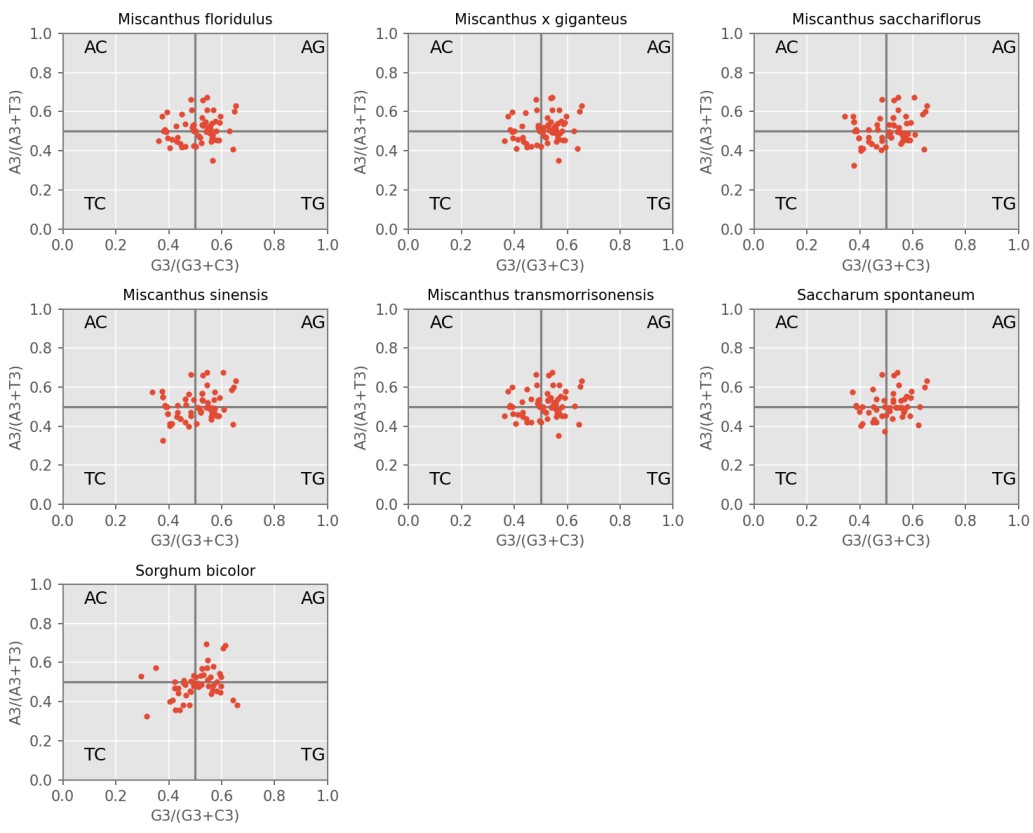

**Figure 3** PR2-plot of chloroplast genomes of seven *Misacanthus* and related species.

1, responsible for ~10% of total variation, was the main source of variation, indicating that the codon usage was influenced by multiple factors. In addition, the relationship between axis 1 and axis 2 was visualized to explore the effects of GC content on codon usage bias (Fig. 5). Genes with different GC content are plotted as different colors, red with GC% < 45% and blue with 45% ≤ GC%< 60% (Fig. 5). In order to determine the factors leading to gene dispersion along axis 1 and axis 2, the correlation index were computed on axis 1 with CAI, CBI, Fop, GC3, GC and L_aa (Table 3). As can be seen from the results in Table 3, axis 1 for *M. floridulus*, *M. sacchariflorus*, *M. sinensis*, *M. transmorrisonensis*, *Saccharum spontaneum* and *Sorghum bicolor* possessed a remarkable correlation with GC3s ($p \leq 0.01$), which indicated the base composition stemming from mutation pressure was the main factor impacting codon usage preference.

## DISCUSSION

The study compared the codon usage patterns of the five *Miscanthus* species and two related species. The findings reported here will help to improve our understanding of evolution analysis and the optimization of codon components suitable for gene expression. During the evolutionary processes, specific codon usage patterns were obtained to adapt to the diverse factors including origin, evolution, natural selection and mutation pressure. In

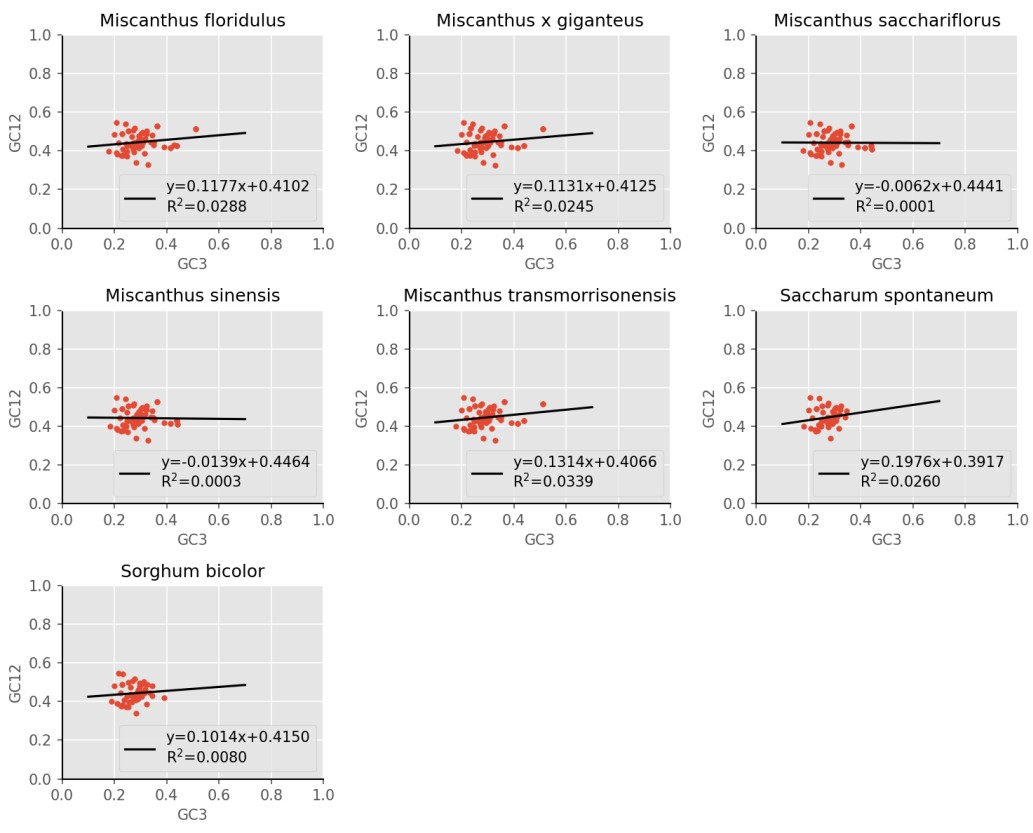

**Figure 4** **Neutrality plot of chloroplast genomes of seven *Misacanthus* and related species.**

addition, analyzing the source of variation in genomic codon usage, the pattern of codon bias and the codon frequency could provide insights into optimization of the codons of heterologous genes and selection of appropriate heterologous expression hosts. Therefore, the research will be of great significance to the study of genetic engineering and genetic evolution.

Our analysis of base composition of codons revealed that the CDSs of the seven *Miscanthus* and related cpDNAs analyzed tended to use an A/T codon, which was consistent with the results of *Zhang et al. (2012)* on the 23 Poaceae cpDNAs that this previous study analyzed. According to a previous study, the GC3 values of dicotyledonous plants is often less than 50% (codon use prefers A/T), which is different from monocotyledonous plants with high GC3 values (GC3 values >50%, showing that codon use prefers G/C) (*Murray, Lotzer & Eberle, 1989*). The results of RSCU value analysis showed an A/T codon usage bias in the cpDNAs of the seven analyzed species, which was consistent with the patterns in most higher plants (*Shang et al., 2011*). According to neutral evolution theory, the effects of mutation pressure and natural selection on the variation of the third base of codon are neutral or nearly neutral (*Sharp et al., 1993*). The study of *Kawabe & Miyashita (2003)* showed that when codon use is affected by natural selection, GC3 values tend to be distributed in a small range and there is no significant correlation between GC12

Sheng et al. (2021), *PeerJ*, DOI 10.7717/peerj.12173

**Table 3** Correlation analysis of axis 1 and codon usage index of chloroplast genomes of seven *Misacanthus* and related species (the $T/C/A/G$ content at the third codon position of synonymous codons; codon adaptation index: CAI; codon bias index: CBI; frequency of optimal codons: Fop; the GC content at the third codon position of synonymous codons: GC3s; the GC content at the three position of synonymous codons: GC; total number of amino acids: L_aa).

| Species | T3s | C3s | A3s | G3s | CAI | CBI | Fop | Nc | GC3s | GC | L_aa |
|---|---|---|---|---|---|---|---|---|---|---|---|
| *Miscanthus floridulus* | −0.65** | 0.507** | −0.133 | 0.598** | −0.152 | 0.073 | 0.098 | 0.28** | 0.639** | 0.159 | −0.315** |
| *Miscanthus x giganteus* | 0.016 | 0.053 | −0.104 | −0.127 | −0.038 | 0.044 | 0.078 | 0.09 | −0.018 | 0.223** | 0.057 |
| *Miscanthus sacchariflorus* | 0.66** | −0.463** | 0.171* | −0.708** | 0.176* | 0.044 | 0.086 | −0.43** | −0.7** | −0.154 | 0.118 |
| *Miscanthus sinensis* | 0.663** | −0.469** | 0.18* | −0.711** | 0.17* | 0.043 | 0.083 | −0.423** | −0.708** | −0.154 | 0.123 |
| *Miscanthus transmorrisonensis* | 0.647** | −0.509** | 0.129 | −0.604** | 0.147 | −0.076 | −0.099 | −0.285** | −0.641** | −0.152 | 0.318** |
| *Saccharum spontaneum* | 0.693** | −0.479** | 0.201* | −0.691** | 0.193 | 0.02 | −0.012 | −0.217* | −0.667** | −0.19 | 0.197* |
| *Sorghum bicolor* | 0.412** | −0.464** | 0.25** | −0.726** | 0.33** | 0.253** | 0.252* | −0.48** | −0.685** | −0.035 | 0.034 |

Notes.
*, $P < 0.05$.; **, $P < 0.01$.
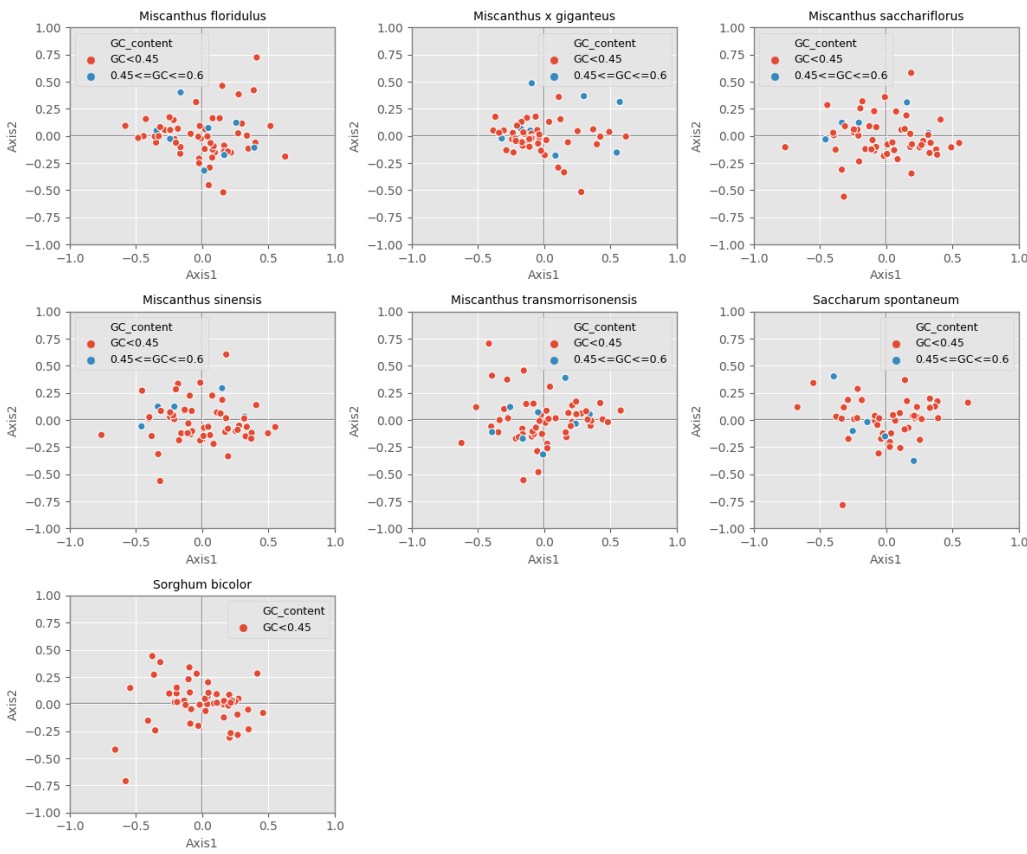

**Figure 5** Correspondence analysis of chloroplast genomes of seven *Misacanthus* and related species.

and GC3 (*Kawabe & Miyashita, 2003*). The neutrality plot in this study revealed a weak correlation between GC12 and GC3 and the composition of the first two bases were different from the third base of the codon, which demonstrated that the codon usage patterns of the seven cpDNAs analyzed are mainly influenced by natural selection. This result is consistent with the codon usage of cpDNAs of many species, such as *Oryza sativa* (*Liu, Tan & Xue, 2003*), *Zea mays* (maize) (*Liu et al., 2010*), *Triticum aestivum* (*Liu & Xue, 2005*) and *Euphorbiaceae* (*Wang et al., 2020*). In addition, combining the results of ENC-plot, PR2-plot and COA analysis suggested that the codon usage bias of the seven cpDNAs were affected by multiple factors, including mutation pressure, base composition and gene length, which the dominant influencing factor was natural selection. This results was consistent with the analysis in *Poaceae* (*Zhang et al., 2012*), *Populus alba* (*Zhou et al., 2008*) and *Euphorbiaceae* (*Wang et al., 2020*) .

The cpDNAs of the seven *Miscanthus* and related plants assessed here are highly conserved and share a total of 21 high-frequency codons. In addition, 4 to 11 codons were determined to be the optimal codons in each species, while no common optimal codon was defined in the seven representative species. These results of high frequency codons and optimal codons are not only beneficial for codon optimization, but also promote further

understanding of the relationship between gene expression and codon usage preference. In higher plants, the main obstacle to applying cp transformation to more species and especially, to other important crops is the limitation of available tissue culture systems and regeneration protocols (*Ruf et al., 2001*). Considering the differences of codon usage bias between the seven cpDNAs analyzed in this study and heterologous expression hosts, codon usage frequencies were analyzed to select the suitable host system. Based on the results, it was suggested to select *Saccharomyces cerevisiae*, *Arabidopsis thaliana* and *Populus trichocarpa* as heterologous expression hosts for *Miscanthus* and related crops, which possesse a little difference in codon usage frequency with the seven plants.

This study conducted a comprehensive comparative analysis on codon usage patterns at the cpDNA-wide level of seven *Miscanthus* and related species. These results will improve our understanding on evolution analysis, the selection of appropriate heterologous expression hosts and the optimization of codon components suitable for gene expression, finally provide a theoretical basis for building a stable and efficient gene expression system in *Miscanthus* or other crops.

## CONCLUSIONS

The codon usage patterns of cp genomes of the five *Miscanthus* and two related species were compared and systematically analyzed for the first time. The results of codon usage bias and RSCU analysis indicated that the seven representative species prefer the use of A/T bases and A/T-ending codons. In addition, 21 common high-frequency codons and four to11 optimal codons were elevated in their frequency in the seven cpDNAs. Furthermore, the analysis of codon usage frequencies between the seven representative species and four model organisms suggested that *Arabidopsis thaliana*, *Populus trichocarpa* and *Saccharomyces cerevisiae* could be considered as appropriate heterologous expression hosts for *Miscanthus* and related species. Finally, when considered together, the results of ENC-plot, PR2-plot and neutrality analysis revealed that the codon usage pattern of the seven cpDNAs analyzed here were influenced by multiple factors, which the dominant influencing factor was natural selection. These results presented in this study potentially provide not only important reference information for evolutionary analysis, but also potentially provide additional important insights for the improvement in the expression efficiency of exogenously introduced genic sequences in transgenic research by codon optimization.

### Funding

This work was financially supported by Natural Science Foundation of the Jiangsu Higher Education Institutions of China (No. 19KJB180023) and Postdoctoral Science Foundation of China (2019M662719). The funders had no role in study design, data collection and analysis, decision to publish, or preparation of the manuscript.

## Grant Disclosures

The following grant information was disclosed by the authors:
Natural Science Foundation of the Jiangsu Higher Education Institutions of China: 19KJB180023.
Postdoctoral Science Foundation of China: 2019M662719.

## Competing Interests

The authors declare there are no competing interests.

## Author Contributions

- Jiajing Sheng performed the experiments, analyzed the data, prepared figures and/or tables, and approved the final draft.
- Xuan She analyzed the data, prepared figures and/or tables, and approved the final draft.
- Xiaoyu Liu and Zhongli Hu conceived and designed the experiments, authored or reviewed drafts of the paper, and approved the final draft.
- Jia Wang performed the experiments, prepared figures and/or tables, and approved the final draft.

## Data Availability

The raw measurements are available in the Supplemental Files.

## Supplemental Information

Supplemental information for this article can be found online at http://dx.doi.org/10.7717/peerj.12173#supplemental-information.

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
