# Peer review of "Comparative analysis of codon usage patterns in chloroplast genomes of five Miscanthus species and related species"

_PeerJ, doi:10.7717/peerj.12173_

## Round 0.1 · original submission · Major Revisions

Dear authors,

As you will see from the 3 reviewer reports, all 3 reviewers are requesting revisions (major) are to be made to your manuscript before resubmission of a significantly improved version of your study.

Please focus on improving the standard of reporting for both your used methodology and the results that you are describing in the manuscript.

English language issues are also raised as a concern by the reviewers and I too share this concern post reading the originally submitted version of your manuscript. Therefore, please have your revised manuscript thoroughly edited by a fluent English language-speaking colleague. If such a colleague is not available to the authorship team, then I strongly suggest that you employ the services of an English language editing service.

I do hope that you consider resubmission of an improved and thoroughly revised version of your originally submitted manuscript for my further consideration of its potential publication in PeerJ.

Kind regards,
Andrew Eamens

Reviewer 1 ·

Basic reporting

The study is original and the manuscript presents relevant data of scientific value. The English language should be improved, mistakes such as those found in lines 27, 29, 171, 186, and 187 make the text less clear and make the reading more difficult. I would recommend you have a colleague who is proficient in English to revise the text or hire a professional editing service. It would improve greatly the quality of the manuscript.
You should describe better the taxonomic context of the chosen studied species. In line 80 it is said that species related to the genera Miscanthus would be included in the study. It would be clearer for the readers to explain the relationship between the species, all of them belonging to the family Poaceae.
When citing a taxon for the first time, please, include information of family and botanical authority: Miscanthus Andersson (Poaceae). Once the species full scientific binomial name was written, you can use abbreviate the genera name, for example, Miscanthus sinensis Andersson for the first time, and from now on in the manuscript, you can refer to this species as M. sinensis. It is worthy to note that species scientific name is always in italic, while species family name and taxonomic authority are never in italic. In line 76 add the scientific name for “Strawberry” following the botanical taxonomic rules. Instead of using “higher plants” (like in line 75), a remnant of scala naturae, use the appropriate nomenclature, Tracheophyta, or vascular plants. The use of dicotyledonous should be avoided as well, instead, you should use eudicotyledons or eudicots (APG et al. 2016).
Please, standardize abbreviation. In line 53 you introduced an abbreviation for Chloroplast as “cp”, but you do not use it consistently throughout the rest of the text. Then, I would recommend you to use it throughout the manuscript or do not to mention it at all. In line 116 you start using ENc (lowercase “c”) to refer to “effective number of codons” but later in the manuscript (lines 145 and 148) you use ENC (uppercase “c”).
The figures, tables, and supplemental data available are informative and high quality. Still, I suggest increasing space between graphs in Figure 2 and in the supplemental tables, instead of using different tabs for each species, perhaps would be easier to visualize if all of them are gathered in the same tab.

Experimental design

The analysis you performed could be better explained. I suggest you justify the choices you made during your research. For example, it would be interesting to explain why you chose those three model species specifically or why you excluded specific codons from your analysis. It could help fellow scientists to learn from your paper and perform similar analyses.
Please, explain that the GC1, GC2, and GC3 refer to the GC content of the first, second, and third nucleotide of each codon, respectively.
The concepts of RSCU and RFSC, as well as its computation, were not clear, try to use the formula to explain it better, as done by Wang et al. 2020 (PeerJ 8:e8251 https://doi.org/10.7717/peerj.8251).
In line 124 you refer to an internet address for a tool you used, however, the page is not available anymore.
It would be very helpful for other scientists if you could add to the supplemental material the python scripts used for this study. It could likely improve the number of accessions and citations of this work.

Validity of the findings

The present manuscript sets clearly what are its aims and shows that your research was able to determine the codon usage patterns of the 7 Poaceae species used in this study as well as provide scientific evidence to suggest the use of the three model species as receptor systems for heterologous gene expressions of Miscanthus.
In the results section, I suggest you revise the species and other taxa nomenclature (as suggested before) and table number in line 300, it seems you mean table 3 instead of table 2.

Additional comments

The manuscript entitled “Comparative analysis of codon usage patterns in chloroplast genomes of five Miscanthus species and related species” wrote by Sheng and collaborators is original and presents relevant data of scientific value that will be useful to broaden our understanding of the evolution of codon usage in the plant lineage. However, the text needs some revision and correction by the authors in order to be published.

Reviewer 2 ·

Basic reporting

The manuscript by Jiajing Sheng and colleagues obtained the chloroplast genomes of five different Miscanthus plants and other two related species from the Genbank database. The values of RSCU and RFSC from seven represent species chloroplast (cp) genome were calculated by Codon W. Then multivariate statistical analysis combined by ENc, ENc-plot, PR2-plot, Neutrality plot analysis was conducted to explore the factors affecting the usage of synonymous codons. Their results showed that the chloroplast genomes of the seven represent species were preference to A/T bases and A/T-ending codons. The codon usage pattern of the seven chloroplast genomes is influenced by multiple factors, which the main influencing factor was nature selection. This study will provide important reference information for evolutionary analysis. However, I have several suggestions as listed below for helping the authors to improve their manuscript.

Experimental design

Rigorous investigation performed to a high technical & ethical standard. Methods described with sufficient detail & information to replicate.

Validity of the findings

Conclusions are well stated, linked to original research question & limited to supporting results.

Additional comments

1. Line 91-94: The complete chloroplast genomes of Miscanthus floridulus (NC_035750.1), Miscanthus sacchariflorus (NC_028720.1), Miscanthus sinensis (NC_028721.1), Miscanthus x giganteus (NC_035753.1), Miscanthus transmorrisonensis (NC_035752.1), Sorghum bicolor (NC_008602), Saccharum spontaneum (NC_034802.1) with gene annotation were downloaded from the NCBI GeneBank database.

The chloroplast (cp) genomes of five different Miscanthus plants were selected as the research objects in this study. However, it is not clear as to why Sorghum bicolor and Saccharum spontaneum were also chosen as the research objects.

2. Line 95-99: To avoid sampling bias, the CDS sequences were screened from genome-wide data by python script according to the following principles: (1) CDS contains initiation codon (ATG), termination codons (TAA, TAG or TGA) and without intermediate stop codon in the sequences; (2) the number of bases in each CDS must be the fold of three (3) the length of sequence of CDS should be≥300 bp.

I believe that the filtering principle of sequence is reasonable. However, I did not find the reliable references for the filtering principle of sequence. I have to doubt whether this principle is rigorous.

3. Line 122-124: The RSCU of each codon was obtained from the sequence files of the high and low groups according to the cusp function of emboss (http://bioweb.pasteur.fr/seqana/interfaces/codonw.html).

The following occurs when the web address (http://bioweb.pasteur.fr/seqana/interfaces/codonw.html) is opened, Cannot GET /seqana/interfaces/codonw.html.

4. Line 131-135: To further explore the codon usage patterns in the seven species of Miscanthus and their relatives, codon usage bias data of four model species including Escherichia coli; Saccharomyces cerevisiae; Populus trichocarpa and Arabidopsis thaliana were downloaded from the Codon Usage Database (http://www.kazusa.or.jp/codon/cgi-bin/showcodon).

The detail of codon usage data of the four model species in this paper is not clear.

5. Line 151-154: PR2-plot is a graphical analysis takes G3/(G3+C3) as the abscisic and A3/(A3+T3) as the ordinate, which is performed to explore the composition of the four bases at the third site of amino acids (Sueoka 1999). The pattern of splashes around the central spot (A=T, C=G) indicate the extent and orientation of the base offset.

The most original literatures about PR2 calculation method are Sueoka 1995 and Sueoka 1999 (Sueoka N. 1995. Intrastrand parity rules of DNA base composition and usage biases of synonymous codons. Mol Evol 40:318-325. Sueoka N. 1999. Translation-coupled violation of Parity Rule 2 in human genes is not the cause of heterogeneity of the DNA G+C content of third codon position. Gene 238:53-58.). The four-codon amino acids, composing alanine, glycine, proline, threonine, valine, arginine (CGA, CGU, CGG, and CGC), leucine (CUA, CUU, CUG, and CUC) and serine (UCA, UCU, UCG, and UCC) (Sueoka 1995), should be considered in the calculation of PR2.

The whole paper is lack of innovation. There is no in-depth expression analysis of codons. In spite of this, I would like to thank the authors and encourage them to improve and complete their work.

·

Basic reporting

The article is a simple and interesting study that investigates codon usage patterns of cp of five Miscanthus and two related species. However, I suggest that a major revision needs to be done before the article is suitable for publication. Listed bellow are some major points that need to be fixed:

1- The English language should be improved to ensure a clear understanding of the text. Some sentences are ambiguous or built using incorrect sentence structure. Typos were also observed. Some examples where the language could be improved are:
Line 54 – please remove “the” before small sizes
Line 72-74 – unclear sentence. Please re-write. Do you mean that the synonymous codon bias have been proven to exist in those cited species? Or the synonymous codon bias is applicable to something else? If so, for what?
109-110 – Sentence is not adequate. Please rephrase it as a possibility.
117 - please replace “reflecting” by “reflects
161 – The word “Meanwhile” doesn’t fit in the sentence
164-166 – Is this already a result or the possibilities? If the last, I would suggest adding an “if” or “when” at the beginning of the sentence.
171- please replace “compares” by “compared”.
206 – please replace “closely” by close

2- Your introduction needs more detail. I recommend using the reference: Plotkin, J., Kudla, G. Synonymous but not the same: the causes and consequences of codon bias. Nat Rev Genet 12, 32–42 (2011). https://doi.org/10.1038/nrg2899 to improve the background information on lines 68-70. I also suggest that you improve the description at lines 79- 86 to provide more justification for your study. Please re-write the paragraph highlighting the aims, objectives or main questions that your work is going to address as well as possible hypothesis that you have.

3- Please improve the caption of the figures so they are self sufficient. Also, Figure 1 needs a title on the X axis.

Experimental design

I liked the idea of performing a study only with data that is already available on a public database. It is an interesting work but as mentioned before I would like to see the research questions clearer defined. I would also recommend inserting a small explanation on why choosing the four model species. There was some ambiguous information that needs to be addressed, for example:
Line 125-127 – Have you calculated the RSCU as described on lines 107-110 or have you obtained them from the files?
Line 164-166 – Is this already a result or the possibilities? If the last, I would suggest adding an “if” or “when” at the beginning of the sentence.

Validity of the findings

The study found interesting results and suggests appropriate exogenous expression receptors for Miscanthus and related species. However, I think that the results and discussion sections need to be restructured. There was lots of information contained in the tables that were repeated along the text. I also noticed that some of the discussion is embedded in the result part. I suggest the authors to include further comparison with other studies in the discussion section as well as clarifying what are the implications (e.g evolutionary consequences) of your findings. For example, in lines 340-342 please cite which are those multiple factors influencing the codon usage bias. You also cite that: "no common optimal codon was defined in the seven represent species", are there any hypothesis on why that occurs? Was that anticipated?

Additional comments

The study has a lot of potential but it needs to be restructured according to the major concerns listed above. There are also minor revisions that require attention:
1. Please pay attention to species names along the text and the reference list - they all need to be in italics.
2. Line 140-148 – Is Enc or ENC the correct abbreviation?
3. Line 59-60 – is there any reference for this sentence? Might be out of context.

---

## Round 0.2 · Minor Revisions

Dear Authors,

Please address the concerns raised by the two peer reviewers of your submitted article - these are largely minor in nature, and therefore, my decision is to request MINOR REVISIONS to be made to your original submission.

I have also reviewed your study and have identified numerous issues which require correction prior to resubmission of your revised article. These can be found in the attached annotated PDF.

Please take your time when carefully considering all the comments and concerns raised in the peer review of your manuscript prior to resubmission of a revised manuscript.

All the best,
Andrew Eamens

Reviewer 1 ·

Basic reporting

The writing quality of this paper has increased greatly since the first submitted version. However, I noticed some inconsistencies in the botanic taxonomic nomenclature, such as genus names not italicized (lines ) or the absence of taxonomic authority (lines ). I recommend checking on the botanical taxonomic nomenclature to correct these minor mistakes.

Experimental design

no comment

Validity of the findings

no comment

Additional comments

no comment

Reviewer 2 ·

Basic reporting

The article must be written in English and must use clear, unambiguous, technically correct text. The article must conform to professional standards of courtesy and expression.

Experimental design

Research question well defined, relevant & meaningful. It is stated how research fills an identified knowledge gap.

Validity of the findings

Impact and novelty not assessed. Meaningful replication encouraged where rationale & benefit to literature is clearly stated.

---

## Round 0.3 · Minor Revisions

Dear authors,

I have previously requested numerous changes to be made to your manuscript which you have failed to address in this version of your manuscript.

In addition, I have identified many additional issues with the latest version of your manuscript which also require attention.

Please make all changes that I am requesting whether you agree with them or not. All changes are required in order for me to consider progressing your manuscript any further through the system,
If you fail to make the requested changes during this phase of the review process then I will have to reconsider the progression of your study any further through the system.

I also require a rebuttal letter where you outline all changes that I am requesting.

Thank you,
Andrew Eamens.

---

## Round 0.4 · accepted · Accept

Dear authors,
Thank you for your continued improvement of your submitted study.
It is my opinion that the manuscript post its multiple rounds of revision is now at the standard suitable for publication acceptance.

Thank you again for following through with the revision process to this stage.

Andrew